# The weakness of fragility index exposed in an analysis of the traumatic brain injury management guidelines: A meta-epidemiological and simulation study

**Thomas M. Condon[1], Richard W. Sexton[2], Adam J. Wells[3,4], Minh-Son To[5,6]***

**1** Department of Intensive Care Medicine, Royal Adelaide Hospital, Adelaide, South Australia, Australia,
**2** Department of Emergency Medicine, Lyell McEwin Hospital, Elizabeth Vale, South Australia, Australia,
**3** Faculty of Health and Medical Sciences, The University of Adelaide, Adelaide, South Australia, Australia,
**4** Department of Neurosurgery, Royal Adelaide Hospital, Adelaide, South Australia, Australia, **5** Division of Surgery and Perioperative Medicine, Flinders Medical Centre, Bedford Park, South Australia, Australia,
**6** College of Medicine and Public Health, Flinders University, Bedford Park, South Australia, Australia

* Minh-Son.To@sa.gov.au

## Abstract

### Objectives

To perform fragility index (FI) analysis on the evidence that forms the basis of the guidelines for the management of severe traumatic brain injury (TBI), and develop a deeper understanding of the pitfalls associated with FI.

### Design

Meta-epidemiological analysis and numerical simulations.

### Methods

The Brain Trauma Foundation guidelines (4th edition) for management of severe TBI were used to identify relevant randomised controlled trials (RCTs). FI based on Fisher's exact test and relative risk was performed on eligible RCTs. The relationship between FI, event counts and *P* values was explored by exhaustively considering different combinations of outcomes for studies of total size ranging from 80 to 10000. Sample size calculations were also performed for a range of power, baseline risk and relative risk, to determine the influence of study design on FI.

### Results

FI analysis of the severe TBI management guidelines revealed that most studies were associated with a low FI. In the majority of studies, FI was of a similar magnitude to the number lost to follow-up. The simulations revealed that while FI was inversely related to *P* value, a wide range of FI may be associated with a given *P* value. FI is also affected by sample size, baseline risk and effect size. Sample size calculations suggest that aside from very high-

**Funding:** Author AJW is supported by the Neurosurgical Research Foundation and is the recipient of the Abbie Simpson Clinical Fellowship.

**Competing interests:** The authors have declared that no competing interests exist.

powered studies, most are likely to yield low FI values in the range typically encountered in the literature.

## Conclusions

Many studies are underpowered and are expected to be associated with a small FI. Furthermore, FI over-simplifies the complex, non-linear relationships between sample size, effect size and *P* value, which hinder comparisons of FI between studies. FI places undue importance on the "significance" of *P* values and accordingly should only be used sparingly.

## Introduction

The concept of fragility was introduced by Feinstein (1990) in the epidemiology literature [1] but has enjoyed a resurgence in the assessment of biomedical research due to its straightforward and intuitive interpretation [2, 3]. The fragility index (FI) represents the minimum number of events required to change the statistical significance of a study result from significant to nonsignificant [2], and has largely been promoted as a measure to assess the robustness of randomised controlled trial (RCT) results with a dichotomous endpoint. Evaluations of FI have been performed in diverse areas ranging from heart failure [4], diabetes [5], nephrology [6], anaesthesiology [7], oncology [8], ischaemic stroke [9], intracranial haemorrhage [10], to paediatrics [11]. These studies have demonstrated that many RCTs have a low FI, with median FI ranging from 2 to 26 (median of the median 4), suggesting that the interpretation of RCT results commonly hinge on a small number of results.

While being promoted as an easy to understand measure, the properties of FI have not been well elucidated. Carter et al. [12] performed simulations of clinical trials and demonstrated that FI is inversely correlated with *P* value and increases with sample size, despite a constant underlying true response rate. Thus, they concluded that FI is a reflection of *P* value and therefore is not an indicator of treatment effect size. A fragility quotient (FQ) has been proposed [13], defined as the FI divided by the total sample size. Although simple, it is not clear that a linear scaling of FI enables direct comparison across studies of different sample size [14]. The use of FI as a measure of robustness is not without limitations either. These include its applicability only to balanced RCTs with two groups and dichotomous outcomes [2]. Furthermore, FI alone does not convey a measure of precision. Therefore, some have argued that FI needs to be interpreted in conjunction with other measures including *P* value, sample size, confidence interval and number lost to follow-up [12–15].

The purpose of this work is to demonstrate the application of FI analysis to the traumatic brain injury (TBI) literature and also to assess the utility of FI as a measure of robustness. Despite considerable effort being dedicated to randomised controlled trials in the study of TBI, the results have proven difficult to translate to clinical practice [16]. The guidelines for the management of severe TBI as published by the Brain Trauma Foundation are now in their 4th edition [17], and encompass a wide-range of management criteria including decompressive craniectomy, steroids and nutrition, hyperosmolar therapy, seizure prophylaxis, and sedation strategies. The guidelines are a careful synthesis of the available published evidence and comprise of five Class 1, 46 Class 2, 136 Class 3 studies, and two meta-analyses. Study designs include a mix of randomised controlled trials (RCTs) as well as observational studies. The guidelines were therefore used to identify RCTs for fragility analysis. The properties of FI were subsequently explored using numerical simulations.

## Methods

Two reviewers (TMC and RWS) independently reviewed abstracts or full-text articles of all identified RCTs from Part 1 (treatments) of the guidelines for the management of severe TBI as published by the Brain Trauma Foundation (4[th] edition) [17]. In order to qualify for inclusion, an RCT had to be of parallel 2-group design (i.e. intervention and control/comparator) with patient randomisation in a balanced 1:1 manner and report at least one dichotomous outcome regardless of statistical significance. In studies reporting multiple suitable outcomes, only one outcome was analysed for fragility, preferably the primary outcome. Before proceeding to data extraction, a consensus was reached on the RCTs meeting the inclusion criteria. Any disagreement was resolved by a third reviewer (M-ST).

Data extraction was performed independently by two reviewers (TMC and RWS). Data extracted included the trial outcome, the number of patients randomised to intervention and control/comparator group, the number who experienced an outcome in the intervention and the control/comparator group, and the number lost to follow-up. Studies were also assessed to determine if power analysis and sample size calculations were performed.

All analyses and simulations were performed using MATLAB 2019b (The MathWorks, Inc.). A custom routine was used to calculate the FI [2]. A two-by-two contingency table of treatment group against outcome (event or non-event) was constructed for each individual study. If the initial Fisher's exact test yielded a "significant" $P$ value (i.e. $P < 0.05$), an event was added to the group with fewer events (and a non-event was subtracted from the same group) and Fisher's exact test was re-calculated. This was iteratively performed until the $P$ value of Fisher's exact test was greater or equal to 0.05. The $FI_F$ was then taken to be the number of events added to achieve a "non-significant" $P$ value. Alternatively, if the initial Fisher's exact test produced a "non-significant" $P$ value (i.e. $P \geq 0.05$), an event was subtracted from the group with fewer events (and a non-event was added to the same group) iteratively until a significant test result was reached. Since relative risk is commonly used to measure effect size in RCTs [18], a similar analysis was performed based on one-sided significance testing of relative risk with a significance level of 0.05, yielding $FI_{RR}$.

### Power and sample size analysis

Hypothetical studies were simulated using a combination of study power ($1-\beta$) ranging from 0.5 to 0.9, baseline risk in the comparator (control) group ranging from 0.05 to 0.4, and relative risk ($R$) ranging from 0.5 to 2.0 (excluding 1.0). There were 152 combinations for each power level, giving a total of 760 combinations investigated. Using these parameters, sample size calculations were performed as described previously [19]. In brief, suppose the true population proportions are $P_1$ and $P_2$. The null and alternative hypotheses being tested are

$$H_0 : P_1 = P_2$$

$$H_1 : \frac{P_1}{P_2} = R \, (\text{for } R \neq 1)$$

The baseline risk (in group 2) is assumed to be known and equal to $P$. The combined sample proportion, $p_c$, is not known but can be approximated by

$$p_c \approx \frac{P(R+1)}{2}$$

Then, for a one-sided test, the total sample size $N$ for a balanced study is given by

$$N = \frac{2}{(R-1)^2 P^2} \left[ z_\propto \sqrt{2p_c(1-p_c)} + z_\beta \sqrt{RP(1-RP) + P(1-P)} \right]^2$$

where $z_\alpha$, $z_\beta$ are critical values of the normal distribution. For a study described by a certain set of parameters, the size of each study group and the expected event counts in each group were obtained under the assumption the alternative hypothesis $H_1$ was true, using the calculated total sample size $N$. The significance level ($\alpha$) was set to 0.05 in all simulations.

## Results

### Characteristics of included trials and outcomes

The review of the TBI guidelines yielded 43 studies that met the inclusion criteria. The studies and the outcomes investigated are listed in S1 Table in S1 File. One included study (Study ID 4 in S1 Table in S1 File) was underway but not yet completed at the time of publication of the guidelines. These studies were enlisted from eight of the 11 topics in the recommendations for treatment interventions in managing severe TBI. No studies from the topics of cerebrospinal fluid drainage, ventilation therapies or deep vein thrombosis prophylaxis met the inclusion criteria. In most topics, a favourable/unfavourable Glasgow Outcome Score or related score (Extended Outcome Score, modified Glasgow Outcome Scale) was the dichotomous outcome used for analysis (30 of 45 studies). For studies relating to infection prophylaxis and seizure prophylaxis, the outcomes utilised were pneumonia/mortality or seizure, respectively.

The summary characteristics of included trials are shown in Table 1. Total sample size ranged from 24 patients to 10008 patients. The number of patients lost to follow-up ranged from 0 to 454 patients, however, four studies did not provide this measure. The breakdown of event counts for each trial group is shown in S2 Table in S1 File. Given these outcomes, Fisher's exact test was performed and the relative risk was also calculated. The corresponding $P$ values are included in the table. The majority of study outcomes were "non-significant" (i.e. $P \geq 0.05$) using either Fisher's exact test (30/43) or relative risk (24/43).

**Table 1. Characteristics of included trials.**

| Characteristic | Median (IQR) | Min | Max |
|---|---|---|---|
| Sample size | 87 (160) | 25 | 10008 |
| Number of events (intervention) | 20 (19) | 3 | 1828 |
| Number of events (comparator) | 19 (16) | 4 | 1728 |
| Lost to follow-up | 0 (9) | 0 | 454 |
| Fragility index (Fisher's exact test) | -3 (8) | -19 | 79 |
| Positive | 3 (6) | 1 | 19 |
| Negative | -6 (4.5) | -19 | -1 |
| Fragility index (relative risk) | -2 (9) | -14 | 92 |
| Positive | 5 (6.5) | 1 | 92 |
| Negative | -5 (2) | -14 | -1 |
| Endpoint $P$ value | **Fisher** | **Relative risk** | |
| ≥0.05 | 30 | 24 | |
| <0.05–0.001 | 4 | 4 | |
| <0.001 | 9 | 15 | |
| Performed sample size calculations | 14 (yes) | 29 (no) | |

## Fragility index analysis of the TBI management guidelines

The fragility index (FI) based on Fisher's exact test or relative risk (one-tailed test) were evaluated for each trial, denoted $FI_F$ and $FI_{RR}$, respectively. For "significant" results, a positive FI was calculated; for "non-significant" results, a negative FI was obtained (see Methods). Although typically in the biomedical literature positive FI is calculated, the concept for negative FI is analogous [10], with the number of results needed to change the interpretation or conclusions of the study, and hence FI is presented here as a continuous result. The distribution of $FI_F$ and $FI_{RR}$ are shown in Fig 1A. Both distributions for $FI_F$ and $FI_{RR}$ were centred near zero, with a median of -3 (range -19 to 79) and -2 (-14 to 92), respectively. Most values also lay within a small range; 24/43 studies had an $FI_F$ value between -5 and 5, while 28/43 had an $FI_{RR}$ value between -5 and 5. We observed $FI_{RR}$ to be greater or equal to $FI_F$ (Fig 1B), consistent with the notion that Fisher's exact test is a conservative test [20]. That is, a greater disparity between event counts needed to be reached for Fisher's exact test to yield a "significant" result. Considering only positive FI values, the median $FI_F$ and $FI_{RR}$ were 3 (range 1 to 79, 14/43 studies) and 5 (range 1 to 92, 19/43 studies), while considering only negative FI values, the median $FI_F$ and $FI_{RR}$ were -6 (range -19 to -1, 29/43 studies) and -5 (range -14 to -1, 24/43 studies).

We did not observe a strong dependence of FI on sample size (Fig 1C). Two studies involving the same cohort of patients had sample sizes one to two orders magnitude greater than the remaining studies. Excluding these outliers (Study ID 23 and 24, in S2 Table in S1 File), the Pearson correlation coefficients between $\log_{10}$(sample size) and FI were -0.094 (95% CI: -0.39 to 0.22) and -0.020 (95%CI: -0.29 to 0.33), for $FI_F$ and $FI_{RR}$, respectively. On the other hand, FI was strongly inversely related to *P* value according to a linear-log relationship (Fig 1D). Excluding the same outliers, the Pearson correlation coefficients between $\log_{10}$(*P* value) and FI were -0.83 (95% CI: -0.91 to -0.71) and -0.92 (95% CI: -0.96 to -0.86), for $FI_F$ and $FI_{RR}$, respectively.

The number of patients lost to follow-up was also extracted, where that information was available (39/43 studies). This distribution is shown in Fig 1E. The number lost to follow-up was greater or equal to the magnitude of $FI_F$ and $FI_{RR}$ in 12/39 and 11/39 studies, respectively (Fig 1F, left); the number lost to follow-up was less than the magnitude of $FI_F$ and $FI_{RR}$ in 27/39 and 28/39 studies, respectively (Fig 1F, right). However, the magnitudes of $FI_F$ and $FI_{RR}$ were no more than 5 more than the number lost to follow-up in 31/43 studies. Thus, in most studies, the FI calculated using either Fisher's exact test or relative risk was a similar magnitude to the number lost to follow-up.

## Analysis of fragility index

To better understand the relationship between FI, *P* value and sample size, we considered all possible combinations of binary outcomes for studies of different sizes. Studies of total size 80, 400, 2000, 10000 were included in the analysis. Each study was divided into two even groups, labelled intervention and comparator (i.e. control). For each study group, the number of events ranged from zero, up to the group size. For each combination of outcomes, Fisher's exact test was performed to determine if there was an association between outcome and study group. This statistical testing allowed contour maps to be generated, demonstrating the contours of different *P* value "levels" for various combinations of outcomes (Fig 2A). With Fisher's exact test known, $FI_F$ could then be calculated for all combinations of outcomes and contour maps of $FI_F$ could be generated in a similar manner (Fig 2B). In this section we considered positive values of $FI_F$ only, although similar principles can be directly related to negative values of $FI_F$. These contour plots demonstrate that for a fixed number of events in one study group, $FI_F$ values vary linearly with the number of events in the other study group. This contrasts with

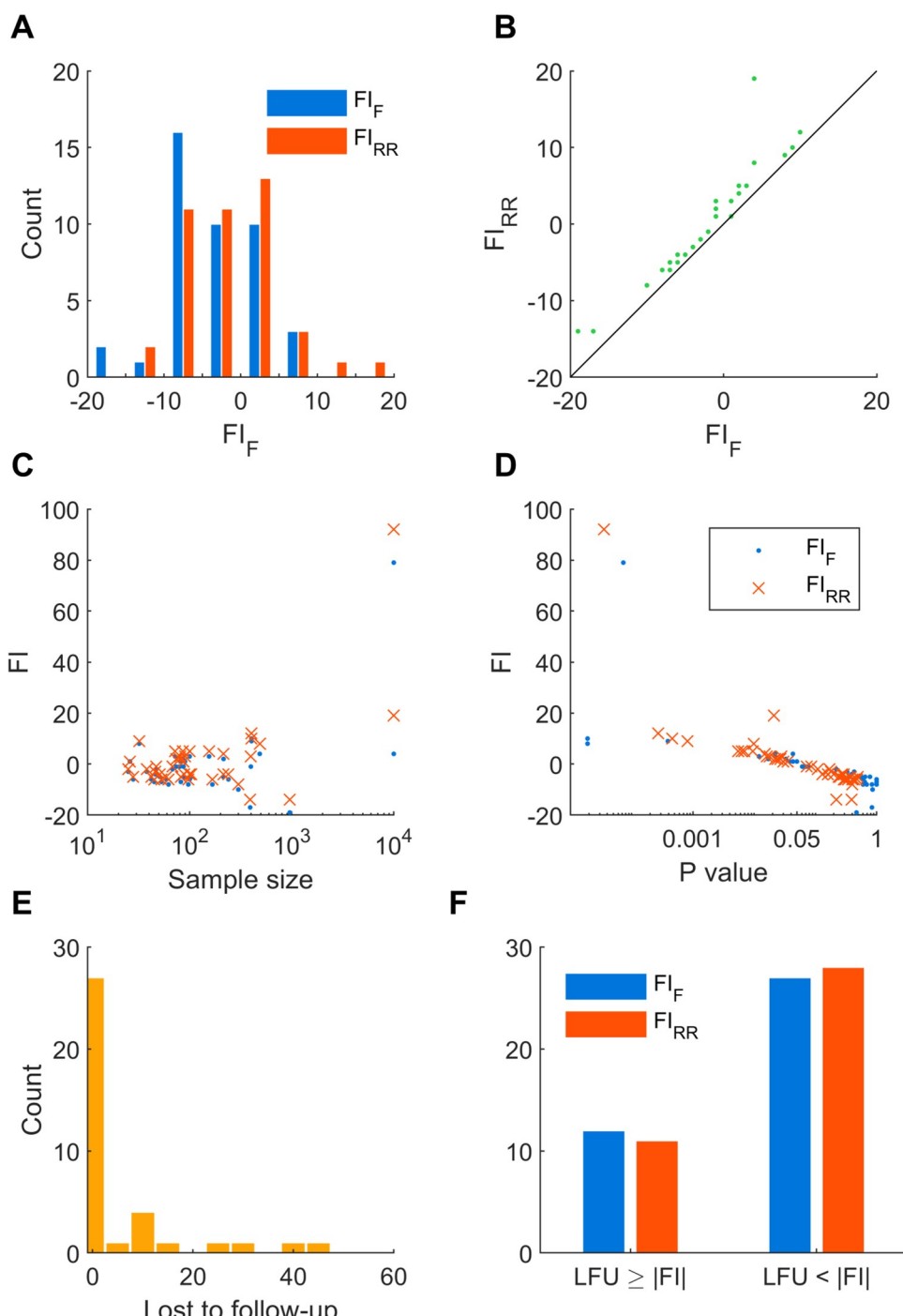

**Fig 1. Fragility index analysis of the TBI guidelines.** (A) Histograms showing the distribution of FI$_F$ (blue) and FI$_{RR}$ (red). One outlier is not shown, corresponding to Study ID 23 (see S2 Table in S1 File) with FI$_F$ and FI$_{RR}$ equal to 79 and 92, respectively. (B) The relationship between FI$_{RR}$ and FI$_F$ is shown. The black line corresponds to equality. Most values for FI$_{RR}$ lie above this line, indicating that FI$_{RR}$ > FI$_F$ for these cases. (C) The relationship between FI (FI$_F$, blue dot; FI$_{RR}$, red cross) and sample size. The horizontal axis is scaled logarithmically. (D) The relationship between FI (FI$_F$, blue dot; FI$_{RR}$, red cross) and *P* value. The horizontal axis is scaled logarithmically. (E) Histogram showing the distribution of the number lost to follow-up. (F) Comparisons of the number lost to follow-up (LFU) and the magnitude of FI.

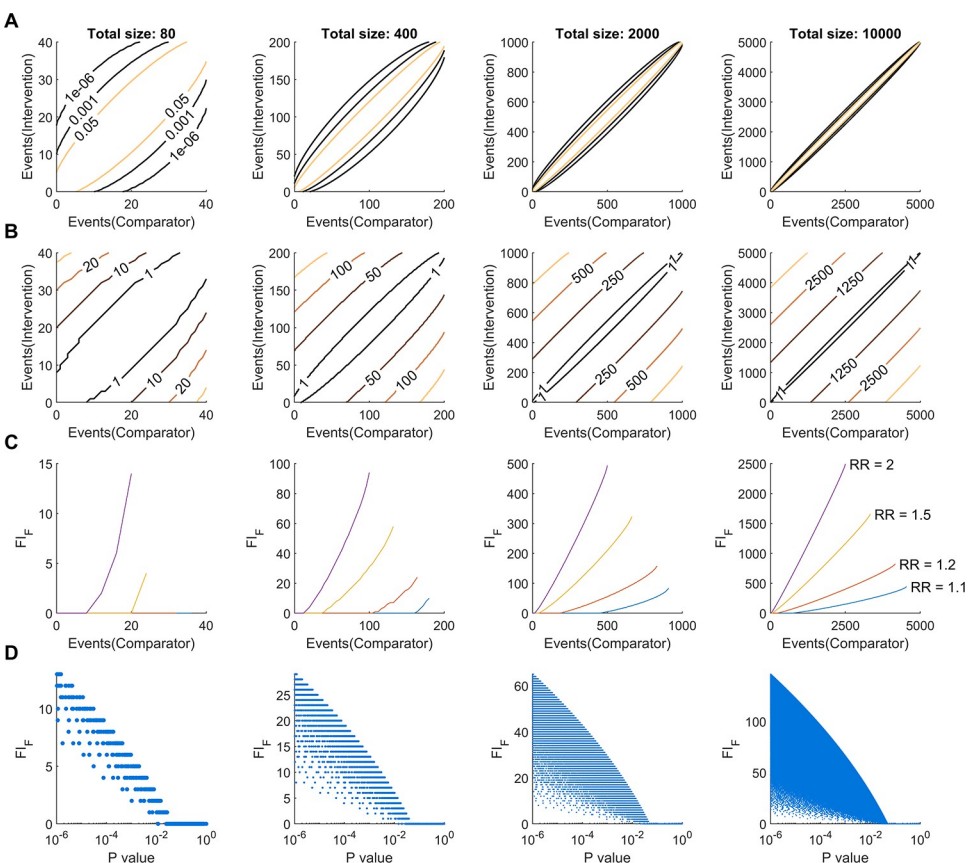

**Fig 2. The relationship between FI, event counts and *P* values.** (A) Contour plots of *P* values for Fisher's exact test. Three *P* value thresholds of 0.05, 0.001 and 1.0 x 10$^{-6}$ were set as contour levels. (B) Corresponding contour plots for FI$_F$. (C) The relation between FI$_F$ and the number of events in the comparator group, for a fixed relative risk. Curves corresponding to a relative risk of 1.1, 1.2, 1.5 and 2.0 are shown. (D) Scatter plots showing the relationship between *P* value and FI$_F$, for $P > 10^{-6}$. Hypothetical studies are balanced, with total size 80, 400, 2000 and 10000.

the contours of *P* values, in that *P* values vary nonlinearly with event counts. Evidently, *P* values rapidly diminish when the disparity in event counts between the study groups widens.

We also explored how FI$_F$ varies with relative risk (RR) (Fig 2C). We considered combinations of event counts that produced the same RR, for four different values of RR; 1.1, 1.2, 1.5 and 2.0. These plots demonstrate that increasing the number of events in the comparator group results in a supra-linear increase in FI$_F$, for positive values of FI$_F$. For values of RR < 1.0, it can similarly be shown that increasing the number of events in the comparator group results in a sub-linear increase in FI$_F$, for positive values of FI. Further, for a given number of events in the comparator group, higher values of RR result in a greater FI$_F$. Furthermore, the relationship between FI$_F$ and *P* values is nonlinear and a given FI$_F$ may be associated with a range of *P* values spanning multiple orders of magnitude (Fig 2D). Taken together, this analysis demonstrates the complex relationships between FI$_F$ and sample size, event counts, relative risk and *P* values.

## Power, sample size and fragility index

Given that FI is related to sample size and significance testing of study outcomes, we then explored if expected FI can be estimated *a priori*, in a similar manner to how power and sample

size calculations are performed as part of study design. Notably, only 14/43 studies reported sample size calculations (Table 1). We considered combinations of a range of study power, baseline risk and relative risk, and generated simulated studies in which the event counts were consistent with the alternative hypothesis (see Methods for further details). $FI_F$ was then calculated based on these event counts and sample sizes.

The results of the simulations are shown in Fig 3. We observed that low powered studies are associated with a lower FI (Fig 3A). For studies with power 0.5, the median $FI_F$ was -3 (range -12 to -2); for studies with power 0.6, the median $FI_F$ was -2 (range -4 to -1); for studies with power 0.7, the median $FI_F$ was 1 (range -2 to 9); for studies with power 0.8, the median $FI_F$ was 4 (range 1 to 25); for studies with power 0.9, the median $FI_F$ was 10 (range 2 to 55). Thus, only some, but not all, higher powered studies were associated with high values for $FI_F$. The range of $FI_F$ also widened with group size, as expected (Fig 3B), but not proportionately. The maximum $FI_F$ observed was 55 for a study of total size 34078, with 1704 and 1874 events in the comparator and intervention arms, respectively. The corresponding simulation

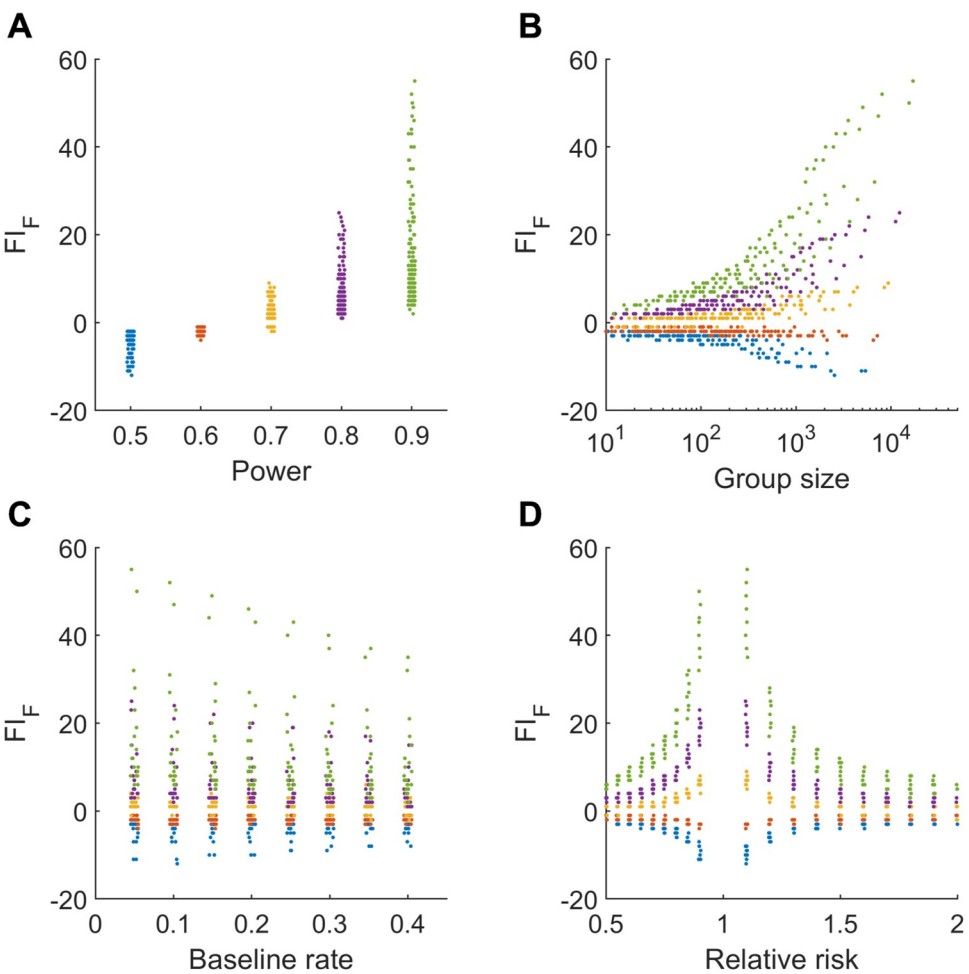

**Fig 3. Sample size calculations and FI in simulated studies.** (A) Scatterplot showing the relationship between FI and study power (1—β). (B) Scatterplot showing the relationship between FI and group size (half the total study size). The group size is shown on a logarithmic axis. (C) Scatterplot showing the relationship between FI and the baseline event rate in the comparator group. (D) Scatterplot showing the relationship between FI and relative risk. Colours represent power levels as in Panel (A).

parameters were study power of 0.9, a baseline risk of 0.05 and a relative risk of 1.10. We also observed that $FI_F$ was only weakly related to baseline risk (Fig 3C). In contrast, higher values of $FI_F$ were realized with relative risk values approaching 1.0, and the demarcation between different power levels was evident (Fig 3D). This is partly explained by the greater sample sizes required to detect a significant relative risk, particularly for higher power levels. Taking all positive $FI_F$ values together, we found the median to be 5 (range 1 to 55; 437 values).

## Discussion

### Fragility index analysis of the TBI literature and comparison with other studies

Fragility index analysis of the BTF guidelines for the management of severe TBI (4th edition) [17] revealed that most studies were associated with a low FI. The trends seen with $FI_{RR}$ were similar to those seen with $FI_F$, although $FI_{RR}$ was typically greater than $FI_F$. This is consistent with the notion that Fisher's exact test is a conservative test for association [20]. Nevertheless, we observed no other advantage in utilising relative risk for FI calculations. We demonstrated that 24/43 studies had an $FI_F$ value between -5 and 5, while 28/43 had an $FI_{RR}$ value between -5 and 5. Furthermore, the magnitudes of $FI_F$ and $FI_{RR}$ was no more than 5 more than the number lost to follow-up in 32/43 studies. We also showed that FI was not strongly related to sample size, suggesting that some of the larger studies hinged on proportionately fewer events. Collectively, these findings suggest that the evidence base supporting the severe TBI management guidelines may be considered "fragile", in that the interpretation of study findings can be swayed by a small number of events. The median positive $FI_F$ was 3, which is similar to other studies that have evaluated FI in different fields [6–9, 11]. Thus our findings are not unique to the severe TBI management guidelines, but appear consistent with the view that much of the biomedical literature is founded on weak evidence [21]. The fragility indices obtained from the BTF guidelines are not being used in this paper to examine the quality of the underlying papers or their clinical implications. Rather, the results are useful in forming a critique of FI itself. We do not advocate for any of these papers or the BTF guidelines themselves to be downgraded in their validity. The challenges of performing RCTs in this population have been well described comprising a heterogenous population with an incompletely understood disease, requiring precise titration of individual therapies. For this reason, RCTs in TBI are often small, expensive and inherently "fragile" [16]. Future research in this field should emphasise sound methodology, large enrolment, patient centred outcomes, and place less weight on *P* values to determine clinical significance. Reporting results with confidence intervals would also provide greater context and relevance upon which to make clinical decisions. Rather than drawing firm conclusions about the severe TBI management guidelines, we discuss below the weaknesses associated with FI.

### Implications on the interpretation and utility of fragility index

Like other areas of clinical research, recruiting participants for TBI management studies can be challenging, and may involve multiple centres in multiple countries over many years. Accordingly, prospective randomised studies typically perform sample size calculations prior to recruitment. These calculations necessarily make assumptions about the expected effect size and population variance, and seek to strike a balance between ensuring the study is sufficiently powered to detect a statistically significant result and over-recruitment of patients [22]. To determine how these considerations may affect FI, we performed sample size calculations for a range of desired power levels, baseline risk and relative risk. The range of FI values realised

was small, up to several orders of magnitude smaller than the study size (Fig 3). Expectedly, underpowered combinations were associated with smaller FI values. It is well known that many RCTs in the biomedical literature are underpowered [23, 24]. It could therefore be said that by design, many studies are inadvertently "fragile". In this light, evaluation of FI may be akin to *post hoc* power analysis, itself an uninformative exercise [25]. Surprisingly however, the median of all the positive FI values in these simulations was 5, and even moderately large, well-powered studies can produce "low" values of FI ($< 20$). Thus, low values of FI are ambiguous. Furthermore, the highest values of FI were only seen in the largest and highest-powered studies To demonstrate this point, the notable outlier in our fragility calculations was the CRASH trial [26] with a $FI_F$ and $FI_{RR}$ of 79 and 92, respectively. With 10,008 participants, it was over ten times larger than the next largest trial [27] with 957 enrolments, and was powered to detect a 2% absolute reduction in mortality [28]. Taken together, these examples highlight key pitfalls with reliance on FI as a measure of robustness.

On face value, FI offers a straightforward interpretation of the susceptibility of a trial to change in the number of events, and this has led to a large number of studies applying FI analysis to various topics in recent years [2–11, 24]. However, such an interpretation overlooks other shortcomings associated with FI. To explore these, we extended our analysis of fragility index with numerical simulations (Fig 2). Our approach is different to that taken by Carter et al. [12] in that we exhaustively considered all possible combinations of event counts for balanced studies of dichotomous outcomes. Although many combinations of event counts may never be realised in clinical trials, this approach enables a complete appreciation of the probability landscape to be obtained. The contour maps demonstrate the essential feature that $FI_F$ varies linearly with event counts. For "significant" results where Events(Intervention) > Events(Comparator), corresponding to the top left regions of panels in Fig 2B, $FI_F$ is simply the horizontal distance to the boundary of significance. Similarly, for "significant" results where Events(Intervention) < Events(Comparator), corresponding to the bottom right regions in Fig 2B, $FI_F$ is the vertical distance to the boundary of significance. These two scenarios are depicted schematically in Fig 4. Since FI is based solely on the boundary of significance, the linear association of FI with event counts will remain regardless of statistical methodology or outcome measure. In contrast, *P* values vary non-linearly (Fig 2A) and consequently, the relation between $FI_F$ and *P* values is highly non-linear (Fig 2D). Although the two values are inversely correlated [12], a wide range of $FI_F$ values may be realised for a given *P* value and vice versa. Thus, $FI_F$ fails to accurately represent the probability landscape for rejecting the null hypothesis in the null hypothesis significance testing (NHST) framework and does not bear a robust relationship to the strength of the evidence.

In isolation, *P* values carry little meaning and are of limited clinical utility. Rather, a *P* value should be reported with an effect size and confidence interval. This notion has previously been detailed in the ASA's statement on *P* values [29] and extensively elsewhere in the scientific literature. In a similar vein, in Fig 2C we show how FI varies with relative risk and event counts. For a given relative risk, smaller event counts in the comparator (or control) group are associated with a smaller FI. Specifically, these results demonstrate that FI is modified by effect size and reinforce the notion that FI should not be considered in isolation. However, the complex relationship between FI and sample size, baseline and relative risks, and *P* value hinders the ability to draw comparisons across different studies.

The dichotomization of *P* values inherent to the calculation of FI epitomises the conventional (albeit arbitrary) selection of the $P = 0.05$ threshold for "statistical significance" in the NHST framework. According to this threshold, a *P* value of 0.049 is considered "significant" while a *P* value of 0.051 is considered "non-significant", despite a nearly indistinguishable Type I error rate. Such *P* values should instead be regarded as providing similarly "weak"

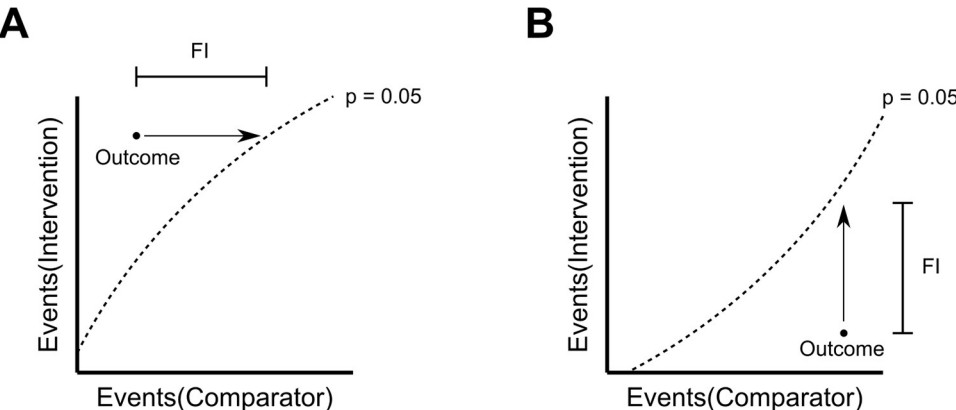

**Fig 4. Schematic description of FI.** (A) In the scenario where Events(Intervention) > Events(Comparator), FI is the horizontal distance to the boundary curve corresponding to $P = 0.05$ (dashed line). (B) In the scenario where Events (Comparator) > Events(Intervention), FI is the vertical distance to the boundary curve corresponding to $P = 0.05$ (dashed line).

evidence for the null hypothesis. Importantly, the dichotomization of $P$ values and emphasis on "statistical significance" has demonstrable effects on the reporting of results in the biomedical literature. Many studies have shown not only a positive skew in the distribution of reported $P$ values but also a large step-change in the $P$ value distribution at 0.05 [30–33]. Such presence of systematic bias in the literature is likely to have far-reaching consequences[21]. Thus, it is our view that FI places undue importance on statistical significance in the interpretation and application of study results.

### Limitations of the study

This study had a number of limitations. The fragility index analysis was restricted only to RCTs from Part 1 (treatment) of the BTF TBI management guidelines and did not encompass the broader TBI literature [16]. No studies from Parts 2 (monitoring) or 3 (threshold) were included. It is also possible that many less significant (or negative) results were not collated into these guidelines. Furthermore, only one outcome per study was considered, typically the primary outcome. Since secondary outcomes are often less powered [34], this may inflate the FI values. The TBI management guidelines also comprise a heterogeneous collection of literature, encompassing medical and surgical, as well as diagnostic and monitoring interventions. Thus, individual studies may not be directly comparable across topics. Finally, our numerical simulations did not take into account the "natural" distribution of parameter ranges found in the literature. Therefore, the distribution of the simulated FI values may not accurately reflect the "true" distribution in the literature.

Notwithstanding these limitations, the FI values obtained in both the analysis of TBI guidelines and the numerical simulations were closely matched to those reported in previous studies [6–9, 11]. This suggests that the range of FI may be relatively insensitive to many of these factors.

### Conclusions

This study is not an indictment on the evidence underlying the published guidelines for management of severe TBI. Rather, we caution the over-zealous use of fragility index as an

indicator of robustness. Power and sample size simulations suggests that many studies are expected to be associated with a small FI *a priori*. Furthermore, the metric over-simplifies the complex, non-linear relationships between sample size, *P* value and effect size and in isolation has limited utility. In combination, it is unclear how much additional information FI provides since it cannot be meaningfully compared across different studies. Neither should FI be taken as a proxy or substitute for *P* value as these are not equivalent. That FI places undue importance on the "significance" of *P* values is in itself hazardous, and perpetuates the fallacy that results are real or not based on an arbitrarily chosen threshold.

## Supporting information

**S1 File.**
(DOCX)

## Acknowledgments

The authors gratefully acknowledge helpful discussions with Dr Shailesh Bihari and Dr Mark Finnis.

## Author Contributions

**Conceptualization:** Thomas M. Condon, Richard W. Sexton, Minh-Son To.

**Data curation:** Thomas M. Condon, Richard W. Sexton.

**Formal analysis:** Minh-Son To.

**Investigation:** Thomas M. Condon, Richard W. Sexton, Minh-Son To.

**Methodology:** Thomas M. Condon, Richard W. Sexton, Minh-Son To.

**Supervision:** Adam J. Wells, Minh-Son To.

**Writing – original draft:** Minh-Son To.

**Writing – review & editing:** Thomas M. Condon, Richard W. Sexton, Adam J. Wells, Minh-Son To.

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
