## [Decision Letter · Decision Letter 0]

22 Jun 2020

PONE-D-20-10984

The weakness of fragility index exposed in an analysis of the traumatic brain injury management guidelines: a meta-epidemiological and simulation study

PLOS ONE

Dear Dr. Minh-Son To,

Thank you for submitting your manuscript to PLOS ONE. After careful consideration, we feel that it has merit but does not fully meet PLOS ONE’s publication criteria as it currently stands. Therefore, we invite you to submit a revised version of the manuscript that addresses the points raised during the review process.

In particular, discussion section should be improved, as pointed out by Reviewer#1. Moreover, the decision to apply the Fragility Index to non-significant findings and to report the "negative Fragility Index" should be clearly motivated in the text.

We look forward to receiving your revised manuscript.

Kind regards,

Laura Pasin

Academic Editor

PLOS ONE

Journal Requirements:

"I have read the journal's policy and the authors of this manuscript have the following competing interests: AJW is supported the by Neurosurgical Research Foundation and is the recipient of the Abbie Simpson Clinical Fellowship"

Reviewers' comments:

Reviewer's Responses to Questions

**Comments to the Author**

1. Is the manuscript technically sound, and do the data support the conclusions?

Reviewer #1: Yes

2. Has the statistical analysis been performed appropriately and rigorously? 

Reviewer #1: Yes

3. Have the authors made all data underlying the findings in their manuscript fully available?

Reviewer #1: Yes

4. Is the manuscript presented in an intelligible fashion and written in standard English?

Reviewer #1: Yes

5. Review Comments to the Author

Reviewer #1: The authors present an elegant and well-written account of the fragility index in TBI evidence included in the BTF Guidelines. The paper is valuable, the simulations are relevant and deserve publication. I do have, however, some comments that need to be addressed:

1. The fragility index was, in essence, meant to be applied to "statistically significant" findings. Spin-offs such as the "reverse FI" have been used for non-inferiority trials. The authors chose to apply the FI to non-significant findings and to report the "negative FI". I would like to ask the authors to motivate their choice methodologically: A non-significant trial with an FI of -2 would have been a significant trial with an FI of 1 had more patients been included who would have had a specific event. Is this something clinically relevant? How should we interpret the "fragility" of non-significant trials? The idea of the FI is that we base clinical practice and policy on trials with low FI, which makes the evidence base on which we so heavily rely on "shaky". Should a fragile non-significant trial influence policy in the view of the authors? In any case, I would like to see the two types of FI reported separately, both in the table and in the text, "FI" and "negative FI" and I think a discussion of this choice and of the meaning of fragile non-significant trials and their impact should be done.

2. The authors do not cite Bragge et al 2016 J Neurotrauma and did not choose to base their calculations on the excellent overview of RCTs in Trauma presented in the respective State-of-Science paper. Can the authors motivate this decision?

3. What conclusions do the authors draw about the set-up of future trials based on their simulations? Can they summarize the advice regarding future trial methodology in the discussion?

4. What are, in the view of the authors, the clinical implications of the FIs in TBI trials? Should recommendations in guidelines be downgraded based on fragile studies? I completely concur with the authors'assessment of the biomedical literature's over-reliance on statistical significance. I personally am in favor of reporting confidence intervals.

5. The authors mention the Fisher test to be a conservative measure of association. They do not mention other methodologies which are used in recent years ( e.g.proportional odds analysis in recent stroke trials or in RESCUE ICP), would this be a solution for more robust conclusions?

6. The CRASH trial is an outlier with a high FI, do the authors consider it a model for future research? Please expand in the discussion

7. Given the difficulties of including patients in TBI RCTs, the heterogeneity of the disease phenotype, the high costs associated with small fragile trials, what would be the solution for the future of TBI evidence generation in the authors'opinion? Please expand in the discussion.

6. PLOS authors have the option to publish the peer review history of their article (what does this mean?). If published, this will include your full peer review and any attached files.

Reviewer #1: Yes: Victor Volovici

---

## [Author Response · Author response to Decision Letter 0]

3 Aug 2020

We thank Reviewer #1 for useful comments. Our responses are detailed below.

Reviewer #1: The authors present an elegant and well-written account of the fragility index in TBI evidence included in the BTF Guidelines. The paper is valuable, the simulations are relevant and deserve publication. I do have, however, some comments that need to be addressed:

1. The fragility index was, in essence, meant to be applied to "statistically significant" findings. Spin-offs such as the "reverse FI" have been used for non-inferiority trials. The authors chose to apply the FI to non-significant findings and to report the "negative FI". I would like to ask the authors to motivate their choice methodologically: A non-significant trial with an FI of -2 would have been a significant trial with an FI of 1 had more patients been included who would have had a specific event. Is this something clinically relevant? How should we interpret the "fragility" of non-significant trials? The idea of the FI is that we base clinical practice and policy on trials with low FI, which makes the evidence base on which we so heavily rely on "shaky". Should a fragile non-significant trial influence policy in the view of the authors? In any case, I would like to see the two types of FI reported separately, both in the table and in the text, "FI" and "negative FI" and I think a discussion of this choice and of the meaning of fragile non-significant trials and their impact should be done.

Our work is focused on the limitations of fragility index and we conclude that fragility index is not a meaningful indicator of robustness. Therefore, we respectfully contend that labelling a significant/non-significant trial as “fragile” is not particularly revealing. 

In the spirit of fragility index, being the number of events that need to change to alter the interpretation of a study outcomes, we included negative FI to demonstrate that both positive FI and negative FI are not meaningful measures. We have made this more explicit in the Results section “Fragility index analysis of the TBI management guidelines” and cited Shen et al (2019, Neurosurg Rev) as a paper that presents negative FI. 

As per the Reviewer’s useful suggestion, we have separated positive and negative FI in the Results section “Fragility index analysis of the TBI management guidelines” and in Table 1. For much of our work, we amalgamated the negative and positive FI in Fig 1 to demonstrate the “continuous” nature of FI. It simply reflects the distance to the threshold of significance (Fig 4). Specifically, most FI values from the TBI studies (both positive and negative) were small in magnitude and fall between -5 and 5 (as we elaborate in the Results). Again, in our simulations, we demonstrate that many study designs will yield a small negative/positive FI (Fig 3). Therefore, we respectfully also argue that separating positive and negative FI does not alter the interpretation that FI is not a useful measure of robustness. The Results section “Analysis of fragility index” concerns only positive FI. 

2. The authors do not cite Bragge et al 2016 J Neurotrauma and did not choose to base their calculations on the excellent overview of RCTs in Trauma presented in the respective State-of-Science paper. Can the authors motivate this decision?

We have now cited this publication, including in the Discussion section “Limitations of the study”. As per our original submission, we elected to use the Brain Trauma Foundation guidelines for management of severe TBI as our source of RCTs. Arguably, fragility index calculations could also have been performed on RCTs presented in Bragge et al (2016). However, this does not alter our interpretation of fragility index, as demonstrated by the simulations, or how we apply this interpretation to the TBI literature. 

3. What conclusions do the authors draw about the set-up of future trials based on their simulations? Can they summarize the advice regarding future trial methodology in the discussion?

The simulations were focused on the properties and pitfalls of fragility index, and demonstrating its limited use in the interpretation of trial results. The simulations were therefore not designed to evaluate study design in general. Although considerations for future trial methodology in TBI research is very important, this is beyond the scope of our work. 

4. What are, in the view of the authors, the clinical implications of the FIs in TBI trials? Should recommendations in guidelines be downgraded based on fragile studies? I completely concur with the authors"assessment of the biomedical literature"s over-reliance on statistical significance. I personally am in favor of reporting confidence intervals.

Due to the limitations associated with the use of FI, we do not think it is appropriate to draw conclusions about the robustness of TBI trials or alter/downgrade the recommendations in the guidelines. We have made this more explicit in the Discussion section “Fragility index analysis of the TBI literature and comparison with other studies”. 

5. The authors mention the Fisher test to be a conservative measure of association. They do not mention other methodologies which are used in recent years ( e.g.proportional odds analysis in recent stroke trials or in RESCUE ICP), would this be a solution for more robust conclusions?

We argue that fragility index itself is not a useful measure of robustness. The choice of statistical methodology/outcome measure does not qualitatively influence fragility index (Figure 4) or overcome its limitations. For example, we have demonstrated this with our calculations of fragility index based on relative risk. While other methodologies (e.g. proportional odds analysis) may yield quantitatively different thresholds for “significance”, any statistical test does not change the notion that fragility index is simply the “horizontal distance” to the threshold (Figure 4). We have elaborated on this in the Discussion section “Implications on the interpretation and utility of fragility index”.

6. The CRASH trial is an outlier with a high FI, do the authors consider it a model for future research? Please expand in the discussion

We have elaborated on the CRASH trial in the Discussion section “Implications on the interpretation and utility of fragility index”. The trial is indeed a notable outlier and over ten times larger than the next largest study included in our analysis. However, as our work is focused on the pitfalls of fragility index, we limit our discussion by mentioning “the highest values of FI were only seen in the largest and highest-powered studies”. We are hesitant to suggest that the CRASH trial should be used as a model for future research (even though it may well be), as there are many other considerations beyond the scope of fragility index analysis that contribute to the quality of research. 

7. Given the difficulties of including patients in TBI RCTs, the heterogeneity of the disease phenotype, the high costs associated with small fragile trials, what would be the solution for the future of TBI evidence generation in the authors"opinion? Please expand in the discussion.

Our work is on the limitations of using fragility index. Our simulations demonstrated even large, well-powered studies can be associated with a “low” fragility index. Our results, taken together, indicate that fragility index is not a useful measure robustness. Thus, we are unable to use it to meaningfully critique the TBI literature. Therefore, evidence generation in TBI research, or indeed other fields of (biomedical) research, should not be altered on the basis of fragility index analyses alone. On account of this, we respectfully argue that offering an opinion on evidence generation is not relevant to our work.

---

## [Editor Report · Decision Letter 1]

5 Aug 2020

The weakness of fragility index exposed in an analysis of the traumatic brain injury management guidelines: a meta-epidemiological and simulation study

PONE-D-20-10984R1

Dear Dr. To,

We’re pleased to inform you that your manuscript has been judged scientifically suitable for publication and will be formally accepted for publication once it meets all outstanding technical requirements.

Kind regards,

Laura Pasin

Academic Editor

PLOS ONE
---

## [Editor Report · Acceptance letter]

6 Aug 2020

PONE-D-20-10984R1 

The weakness of fragility index exposed in an analysis of the traumatic brain injury management guidelines: a meta-epidemiological and simulation study 

Dear Dr. To:

I"m pleased to inform you that your manuscript has been deemed suitable for publication in PLOS ONE. Congratulations! Your manuscript is now with our production department. 

If your institution or institutions have a press office, please let them know about your upcoming paper now to help maximize its impact. If they"ll be preparing press materials, please inform our press team within the next 48 hours. Your manuscript will remain under strict press embargo until 2 pm Eastern Time on the date of publication. For more information please contact onepress@plos.org.

Kind regards, 

on behalf of

Dr. Laura Pasin 

Academic Editor

PLOS ONE